Classification of high-voltage power line structures in low density ALS data acquired over broad non-urban areas

Roussel Jean-Romain 1 jean-romain.roussel.1@ulaval.ca
http://orcid.org/0000-0003-0118-1651 Achim Alexis 1
http://orcid.org/0000-0002-5146-4476 Auty David 2
1 Centre de Recherche Sur les Matériaux Renouvelables, Département des Sciences du Bois et de la Forêt, Université Laval , Québec , Canada
2 School of Forestry, Northern Arizona University , Flagstaff, Arizona , United States of America
Gu Chenghong
Electronic publication date: 2021 Aug 31
Publication date: 2021
Volume: 7
Electronic Location ID: e672
Received 2021 Feb 12; Accepted 2021 Jul 21
Copyright: © 2021 Roussel et al.
Copyright year: 2021
Copyright holder: Roussel et al.
License: This is an open access article distributed under the terms of the Creative Commons Attribution License, which permits unrestricted use, distribution, reproduction and adaptation in any medium and for any purpose provided that it is properly attributed. For attribution, the original author(s), title, publication source (PeerJ Computer Science) and either DOI or URL of the article must be cited.
License URL: https://creativecommons.org/licenses/by/4.0/

Keywords: LiDAR, Powerlines, Segmentation, Transmission towers, Classification, Forest

Funding: Ministère des Forêts, de la Faune et des Parcs du Québec The Ministére des Forêts, de la Faune et des Parcs du Québec This work was supported by the Ministère des Forêts, de la Faune et des Parcs du Québec. The Ministére des Forêts, de la Faune et des Parcs du Québec acquired the data we used in this study.

==============================
Airborne laser scanning (ALS) has gained importance over recent decades for multiple uses related to the cartography of landscapes. Processing ALS data over large areas for forest resource estimation and ecological assessments requires efficient algorithms to filter out some points from the raw data and remove human-made structures that would otherwise be mistaken for natural objects. In this paper, we describe an algorithm developed for the segmentation and cleaning of electrical network facilities in low density (2.5 to 13 points/m2) ALS point clouds. The algorithm was designed to identify transmission towers, conductor wires and earth wires from high-voltage power lines in natural landscapes. The method is based on two priors i.e. (1) the availability of a map of the high-voltage power lines across the area of interest and (2) knowledge of the type of transmission towers that hold the conductors along a given power line. It was tested on a network totalling 200 km of wires supported by 415 transmission towers with diverse topographies and topologies with an accuracy of 98.6%. This work will help further the automated detection capacity of power line structures, which had previously been limited to high density point clouds in small, urbanised areas. The method is open-source and available online.

Introduction

Airborne laser scanning (ALS) has gained importance in recent decades for multiple uses related to the cartography of the earth’s surface. Active airborne LiDAR systems directly capture the 3D information of surrounding objects and generate highly accurate georeferenced 3D point clouds that describe the structure of the land. While ALS first provided a way to build extremely accurate digital terrain models (Nelson, 2013), several other applications have also been developed for predicting and mapping various characteristics of the vegetation and many other features of interest to disciplines such as forestry, ecology and land management. These include the characterisation of wildlife habitat (e.g. Goetz et al., 2007), water bodies (e.g. Canaz et al., 2015; Morsy, 2017) and forest roads (e.g. Ferraz, Mallet & Chehata, 2016), among many others. In forestry, a substantial amount of work has focused, based on documented best practices (White et al., 2013), on the use of ALS data to map forest resources and help forest practitioners make optimised decisions (e.g. Li et al., 2012; Bouvier et al., 2015; Blanchette et al., 2015; Tompalski et al., 2016).

Processing ALS data over areas covering hundreds of thousands of square kilometres for forest resource estimation and ecological assessments requires algorithms to filter out some points from the raw data and remove human-made structures that would otherwise be mistaken for natural objects. Resource estimation in broad coverage is generally performed using the area-based approach (ABA) (Nilsson, 1996; Næsset, 1997; White et al., 2013; Luther et al., 2014; Bouvier et al., 2015). Under ABA, all points are treated equally to derive metrics summarizing their distribution within individual pixels (typically 20 × 20 m) (Tompalski et al., 2016). The typical metrics used to capture and summarize the vertical distribution of points are height-derived, with the premise that higher points belong to taller trees. In the absence of recorded attributes allowing for selective omission of some points from the analysis, those belonging to human-made structures may be incorporated unknowingly as if they were part of the vegetation. Being tall structures commonly found in natural landscapes, the occurrence of transmission towers and wire conductors may thus generate a significant source of error in ABA models. First, transmission towers may be misinterpreted as very large trees, which could introduce bias to biomass estimation models, for example. Second, electrical wires may statistically ‘hide’ vegetation underneath that may be of ecological interest. In such cases, buffering out the known X and Y positions of the electrical distribution network is not a suitable solution. Classification methods are thus required to facilitate analyses describing the vegetation located either directly underneath or in the vicinity of power lines.

Existing methods to discriminate points belonging to power supply structures can be classified into two main types i.e. line-shape-based detection, and machine learning, with some instances of overlap between the two methods. Line-shape-based detection consists of analysing the structure of the point cloud in a tight neighbourhood around each point (Matikainen et al., 2016) to estimate the local shape of objects. Such methods are often based on an eigenvalue decomposition to segment linear features (McLaughlin, 2006; Jwa, Sohn & Kim, 2009; Kim & Sohn, 2010; Ni, Lin & Zhang, 2017; Qin et al., 2017; Chen et al., 2018). For a given point and its neighbourhood, and supposing the computed eigenvalues are defined as λ0, λ1 and λ2, linear features can be identified in a point cloud using the ratio of λ0 to λ1 and λ2 (Jwa, Sohn & Kim, 2009). Other methods based on a Hough transform have also been reported (Melzer & Briese, 2004; Liu et al., 2009). Alternatively, Liang et al. (2011) selected initial seeds manually and then used a region growing method to segment power lines. These line-shape-based methods have been successfully applied to electrical wire detection, with reported accuracy ranging from 87% to 97%. However, they can not be applied to nonlinear features such as transmission towers.

There has been a growing interest in recent research in the use of machine learning approaches to classify point cloud scenes, including the use of random forest classifiers (Kim & Sohn, 2010, 2012; Ni, Lin & Zhang, 2017), convolutional neural networks Zhang et al. (2019), graph convolutional neural networks Li et al. (2020) and Wen et al. (2021), Latent Dirichlet Allocation Yang & Kang (2018), or a hierarchical unsupervised method Wang et al. (2017). Applications are not restricted to power supply infrastructure, but for transmission towers and power lines high levels of accuracy were achieved, with accuracy ranging from 80% to 99%.

Line-shape-based methods are simpler to implement and appear to perform very well in high density point clouds. For example, we have used a simple eigenvalue decomposition method to produce Fig. 1 where a perfect segmentation of the wire conductors was obtained with minimal programming effort and without any prior information or supervision on the location of the tower or the orientation of the wires. In practice, however, such a method has important limitations, especially in broad, non-urban and low-density coverage. The method assumes wires are continuous structures, but this assumption cannot hold in low-density point clouds, which are typically acquired over vast landscapes. In such datasets, power lines present several gaps with sparsely distributed points (Fig. 2), thus making analyses of the local geometry unfeasible.

Figure 1 Wire conductor segmentation using an eigenvalues decomposition of the k-nearest neighbourhood of each point to evaluate the elongation of the local point structure.

The method works very well without any prior information on the network, as long as the wires are sampled homogeneously and continuously.

Figure 2 3D rendering of classified point clouds on a digital terrain model (to scale) representative of the reality of broad natural landscape coverage.

Towers (blue) and wires (red) for four selected subsets are shown. In all figures, the classification is perfect (no false positives or false negatives) but in (A) conductors are linear and continuous structures evenly sampled with no gaps so they could easily be segmented based on geometric analysis. In (B) conductors are linear continuous structures with a few discontinuities (gaps). In (C) conductors are linear discontinuous structures partially sampled, while in (D) conductors are sparsely sampled with more missing parts than sampled parts, and even the towers are extremely sparsely sampled so it is only possible to distinguish hanging insulators.

Machine learning methods, on the other hand, may be more robust. However, most reported studies were focused on urban scenes, with very high point densities and no evidence that they could be applied to low-density point clouds resulting from sparse structure sampling, such as those reported in Fig. 2. Even if applicable to such situations, existing models will at least require training with new data. Despite machine learning appearing capable of providing highly accurate semantic segmentation, and even instance segmentation, in high-density point clouds where objects are well-defined and densely sampled, there remain doubts as to the performance of such methods in some sparsely sampled areas of low-density point clouds where transmission towers may be reduced to insulators and wires to sparse outliers, as shown in Fig. 2D.

The aforementioned studies were not constrained by the problem of containing objects barely recognizable to the human eye because point clouds were acquired for the specific purpose of power supply segmentation and monitoring using very high point densities (from 25 to 60 points/m2), and a low flight path often parallel to the power lines (e.g. McLaughlin, 2006; Kim & Sohn, 2013; Chen et al., 2018), or even using a dedicated system (e.g. Qin et al., 2017). Some reported line-shape-based methods are, however, likely more robust to this issue. For example, Jwa & Sohn (2012), McLaughlin (2006) used a three-parameter hyperbolic cosine equation that approximates a catenary curve to regress the points belonging to a wire, which was consequently more robust to gaps and sparse sampling. McLaughlin (2006) achieved an accuracy of ≈87% on wire segmentation in a 2.5 points/m2 point-cloud while Jwa & Sohn (2012) reported 75% but with 24 points/m2 and in more complex scenes.

Despite the diversity and high accuracy of power line segmentation methods described in the literature, none applied to the context of this study. In addition to differences in point cloud densities and the urban vs forested context, previous studies have been designed and tested on much smaller datasets (640 m of power lines for Jwa & Sohn (2012), 2 km for Jwa, Sohn & Kim (2009) and Munir, Awrangjeb & Stantic (2019), 750 m for Sohn, Jwa & Kim (2012), 4 km for Guo et al. (2019), 1 to 2 km for Melzer & Briese (2004) and Zhu & Hyyppä (2014), 10 km for Yang & Kang (2018)), often with linear network topologies with no network forking and few deflections. Finally, none of the existing studies we have reviewed provided access to software for users to implement the method or at least access the source code. Several studies were also described in short proceedings papers with insufficient detail to derive our own implementation of the methods. According to Matikainen et al. (2016), multiple patents have been registered but so far the availability of such tools has eluded the research community.

In this context, there is a need for ready-to-use software that allows forest researchers or practitioners to accurately classify and remove power supply structures in ALS point clouds. We present a method tested on ≈200 km of power lines, across various terrains, tower types, and topologies such as linear networks, but also on the deflections and forks that characterise the structure of transmission towers, and in low-density point clouds with extremely partial sampling of the features of interest, as shown in Fig. 2D. Our method, which was inspired from the previous efforts mentioned above (especially McLaughlin (2006)), takes advantage of an existing map of the power distribution network as prior information. Unlike small power supply facilities in towns that carry energy to end customers, maps of high-voltage power lines are necessarily available and maintained by power supply companies. We also take advantage of the fact that information about the type of transmission towers that hold the conductor wires along a given power line is also available, and is usually invariant along hundreds of kilometres. Our proposed method is fully open-source, reproducible, documented and ready-to-use. It is implemented in R (R Core Team, 2021) and takes advantages of the lidR package (Roussel & Auty, 2021; Roussel et al., 2020) to facilitate processing over broad areas.

Methods

Our classification method relies on two priors: (1) a map of the high-voltage power lines across the area of interest, and (2) knowledge of the type of transmission towers along a given power line. The first prior is trivial as it simply helps focus on areas of interest where the electrical distribution network is located. The second prior implies querying a database in which the specifications of the towers are used to anticipate the sizes and shapes of the towers we are looking for in a given region of interest, as well as the number of wire conductors, how the wires are handled and arranged, etc. Figure 3 shows, for two tower types, how a tower is described in our software. Then, the point cloud segmentation method consists of four main steps: (1) tower tracking to find and map the positions of the transmission towers, (2) topology reconstruction to match consecutive, interconnected towers, (3) tower segmentation in which groups of points are identified as belonging to a transmission tower, and (4) wire segmentation in which groups of points are identified as belonging to an electrical wire. Each of these steps is described in further detail in the following sections.

Figure 3 Description of two types of towers in the database.

All measurements are given within a range of validity because towers of the same type may differ (mainly, but not only, in height). Both types are not to scale, the double circuit type usually being taller.

Step 1: tower tracking

The first step consists of mapping the positions of the transmission towers. Starting from the map of the network in the form of spatial lines, we decompose the network into linear sections and then each line is buffered to encompass the power lines (Fig. 4A). This allows for working iteratively on linear sections with a known orientation and is helpful for reducing scene complexity. This way, the algorithm can focus on smaller numbers of points instead of screening the whole scene, which in our case usually covered an area greater than 1 km2 and was free of any human structures.

Figure 4 Main steps of the transmission tower tracking and topology reconstruction displayed on a 4-km2 region of interest for a difficult case were the network forks.

(A) Original map of the network split in two linear sections in red and black and the associated buffers with dotted lines; (B) tower candidates correction step with a tower incorrectly positioned, and a false positive because an earth wire point was detected as a local maximum; (C) map of the transmission towers found including their bounding boxes and the orientation of the wires. Red towers are those found twice, once per linear section processed; (D) topology reconstruction to retrieve how towers are connected. Light colours correspond to a network reconstructed with virtual towers because there are no towers beyond the limits of the dataset.

In each region of interest we applied a point-cloud-based local maximum filter (LMF) to find the local highest points. The local maximum filter we used is analogous to that normally used to detect individual trees (Popescu & Wynne, 2004), but specifically uses narrow but very long rectangular search windows oriented parallel to the wires to reduce the number of false positives. By design, the LMF usually finds all the towers but tends to also produce several false positives corresponding to high vegetation or to sparse earth wire points that are locally higher than neighbouring points (Fig. 4B). In addition, the true positive towers are usually not very accurately positioned because the LMF finds the single highest point of each tower i.e. the “ears” of the towers instead of the centre (see waist type in Figs. 3 and 4B). The algorithm then fixes these two issues by removing locations that do not actually correspond to a tower position and centres the position of the tower top for the remaining true positives.

Because the success of steps 2, 3 and 4 is highly dependent on the accuracy of step 1, this correction task is critical. If a single tower is missing, or a single false positive tower remains, it will invalidate subsequent analyses. Consequently, a very robust trimming of false positives is key to the success of the segmentation. To achieve this, we extract the surrounding point cloud at the position of each candidate tower (Fig. 4B). The subset is then analysed to confirm whether or not it actually contains a tower. This is based on an analysis of the vertical and horizontal distribution of the points. Interested readers may consult the publicly available source code for further details of how this task is performed. Here, we opted to provide limited detail because this step is likely to be modified and improved in the future, which could in turn invalidate the description provided in this paper.

Using the subset of points surrounding the position of the tower, the realignment of its position is made by averaging the X and Y coordinates of the points located 5 to 10 m below the location of the highest point (Fig. 4B). Because towers are vertically elongated structures, this task is easy to implement and provides good results.

The xyz positions of each tower is then assigned as their centred tops. With the map of the network and the prior information about the tower types, we can also assign the general orientation of the power lines as well as the bounding box of the towers (Fig. 4C). In Fig. 4C the towers drawn in red were found twice in each linear section because the network is forking. These correspond to “deflection towers”, which have wires with different orientations on each side. In Fig. 4C, we can see that two deflection towers were detected, but one of them was a false positive. This is not an issue as will be made evident in the next section.

Step 2: topology reconstruction

Once transmission towers are accurately identified and positioned, step 2 consists of retrieving the network topology by determining how towers are interconnected. For this we use the known orientation of the power line and send 500-m beams in the direction of the previous and next towers with a range of angles corresponding approximately to the width of one transmission tower. The topology of the network is built progressively by identifying the beams that intercept the exact location of the next tower. The 500-m limit was used to avoid scenarios whereby implausibly distant towers are connected by mistake. Figure 4D provides an example of connections in which we retrieved three rows of power lines with one being deflected in a different direction. We can see that the false positive deflection tower presented in Fig. 4C was not an issue because it could not connect with any other tower in the direction of the deflection.

In this step we also needed to introduce the concept of “virtual towers”. Because two towers are needed to compute a connection, we necessarily had troubles at the edges of our datasets where the last towers could not be connected. Virtual towers are thus added when a tower cannot be matched with another tower. While this was always the case at the edges of the studied area, the concept was also useful in cases where towers were missed (false negatives). In such cases, two consecutive towers may be too far apart to be matched, thus creating a gap in the network. The addition of virtual towers helped make the topology reconstruction more robust. The exact location of missing towers was not known, but this was not our purpose. The objectives pursued with the creation of virtual towers were twofold: (1) to ensure the topological validity of networks, and (2) to reduce the effects of missing towers on the detection of wires. The latter will be made clearer after “Step 4: Wire Classification”. Although they are an approximation, virtual towers contribute to reducing the effects of errors made in step 1. There is no missing tower in Fig. 4D, but we can see the prolongation of the network in lighter colours after the last towers at the edges of the area.

Step 3: tower classification

At this stage we know the positions of the towers, their height, their orientation, their type and their interconnection in the network. However, the point cloud remains to be classified. As we know the tower type, we also know their xy dimensions. In this step, we assign all points located within the bounding boxes of the towers down to the ground level as belonging to a transmission tower (Figs. 5A and 5B).

Figure 5 Main classification steps.

(A) Large red points are the transmission tower positions found in step 1 and are known to be connected following the application of step 2. (B) Using the bounding box of each tower centred on the tower position, we classify all points from top to ground as belonging to a tower (in blue). (C) Reconstruction of the catenary curve in purple. (D) Moving this curve below the wires. (E) Classify the points above the curve as wires, including the earth wires that typically look like outliers.

Step 4: wire classification

In this step, we use a well-known phenomenon from physics and geometry to classify points belonging to the conducting wires. An idealized hanging chain or cable supported only at its ends assumes, under its own weight, a type of curve called a catenary. The general equation describing the catenary between two points is (Hatibovic, 2014):

(1) y(x)=2c(A(x)−B(x))+h1

With

(2) A(x)=sinh2⁡(12c(x−S2+c×arcsinh(h2−h12c×sinh⁡(S2c))))

(3) B(x)=sinh2⁡(12(S2c−arcsinh(h2−h12c×sinh⁡(S2c))))

Where (x1, h1) and (x2, h2) are the position and elevation of the cable ends, c a constant that (roughly) corresponds to the tension in the wire, and S the span length i.e. |x2 − x1|.

Despite the apparent complexity of this equation, it has a single parameter c, which is part of the description of the tower, assuming that for a given tower type the tension is always the same. We can thus reconstruct the wire curve from the top of the towers (Fig. 5C). This curve is positioned above the wire, but since we know the tower type we also know the typical distance between the top of the tower and the wires (Fig. 3). We then move this curve below the wires (Fig. 5D) and classify every point above the curve that is not already classified as a tower, as a wire. This includes the earth wires that, in practice, look like noise or outliers but are actually wires (Fig. 5E).

The final classification for the dataset used to describe the method is showed in Fig. 6. Figure 2 was also the result of using this method.

Figure 6 3D rendering of points classified as part of the electrical network on a 4 km2, true scale, digital terrain model.

Datasets

To validate the algorithm, we selected 130 1-km2 tiles containing wires among a 30,000-km2 dataset located in the Côte-Nord region of eastern Quebec, Canada (Fig. 7). Although the tile selection was blind (i.e. made without looking at the point cloud), it was not fully randomised. Tiles were grouped into 28 regions of interest, which were selected manually to include difficult cases, including deflections, forks, gaps, single rows, multiple rows, multiple tower types, etc. A fully randomised selection would have very likely produced only linear sections with waist-type transmission towers, which are the most common in the network. Figure 2 shows some small linear subsets from some of the chosen regions of interest. Our validation dataset consisted of 28 regions of interest, with conductor lengths ranging from 2 to 12 km in areas ranging from 4 to 8 km2. Each section contains an electrical network in single (Fig. 2A) or multiple rows (Fig. 2B). To estimate the network length, we multiplied the length of the power lines by the number of rows. For example in Fig. 2B there are three times 900 m of wire conductors i.e 2.7 km.

Figure 7 Location of the 3,0000 km2 dataset in the Côte-Nord region of Quebec, Canada.

The 28 regions shown in red were used to validate the algorithm.

The point density averaged 7 points/m2 over the full dataset and ranged locally from 2.6 to 13.8 points/m2. The full 30,000-km2 dataset was not sampled in a single contract was acquired over several years. Thus, the densities tended to change among regions. Because of the multiple contractors, the acquisition devices also varied. For these reasons, the sampling design specifications are not reported here. Our proposed method should be seen as working independently from the acquisition device and point density.

Validation

To evaluate the accuracy of the method, we manually assessed the results obtained in our 28 selected regions of interest. In the absence of reference data on the positions of the transmission towers, we manually assessed the tower segmentation and counted the number of true positives, false negatives and false positives. This relatively trivial task, was performed on a total of 415 towers.

Assessing the accuracy of the wire segmentation was more challenging. In the absence of a perfect segmentation upstream of the method, we could not directly and automatically estimate the percentage of points that were correctly or incorrectly classified. The key to the successful application and development of our method was to identify situations in which points tend to be misclassified. To this end, counting the percentage of correctly classified points did not bear much relevance. For example, we could miss all points belonging to conductor wires in dataset #15 (for which a subset is presented in Fig. 2D) without significantly affecting the overall percentage of correct classification because this dataset is so sparsely populated that it actually accounts for a very small proportion of the points included in our analysis. We thus chose the length of true positive, false positive and false negative sections of power supply facilities as a quantitative measure of segmentation accuracy. This way, the same weight was attributed to each dataset independently of their sampling density. By design, the misclassification errors output by our algorithm are grouped into continuous clusters. Results were thus free of isolated and randomly-located misclassified points, which ensures the relevance of the chosen metric. This approach also allowed us to derive some key statistics to assess model performance, such as precision, recall and F-score, which were interpreted in kilometres of power lines instead of percentage of points.

Results

Accuracy

Table 1 shows, for the 28 evaluation datasets, the dataset IDs (so we can refer to them in the text), the number of power line rows, the number and types of transmission towers supporting the wires, as well as the number of true positives, false positives and false negatives. Regarding the wires, the table also presents their total length (i.e. the length of the network multiplied by the number of wire rows) as well as the total lengths of true positive, false positive and false negative sections. Overall, 21 of the 28 trials were perfectly classified with with 0 false positives and 0 false negatives. Figure 6 provides an example of such perfect classification obtained in dataset #13. The seven remaining datasets contained various degrees and types of classification errors. The classification errors of datasets #1 #15 and #17 are presented in further detail in the next sections to highlight some limitations of the method. The NA values associated with datasets #10 and #22 correspond to areas where we considered the algorithm had failed. In both cases, largely incorrect results were produced whereby false positives and false negatives were no longer clustered in continuous sections.

Table 1 Summary of the results for our 28 evaluation datasets containing from 2 to 12 km of wires.

dc, double circuit; wt, waist-type; wts, waist-type-small which look similar to waist-type but with smaller towers. Accuracy for tower segmentation is measured with number of towers missed or falsely detected. Accuracy for wire segmentation is measured with length of wire missed or falsely detected.

ID	Rows	Towers (unitless)	Wire (km)	
		Type	n	TP	FP	FN	Length	TP	FP	FN	
#1	2	dc	32	29	0	3	12.14	11.4	0	0.74	
#2	2	dc	32	32	0	0	12.4	12.4	0	0	
#3	1	wt	13	13	0	0	6	6	0	0	
#4	1	wt	12	12	0	0	6	6	0	0	
#5	2	wt	24	24	0	0	10	10	0	0	
#6	2	wt	26	26	0	0	11.2	11.2	0	0	
#7	1	wt	11	10	0	1	5.7	5.55	0	0.15	
#8	4	wt	39	39	0	0	17.2	17.2	0	0	
#9	3	wt	32	32	0	0	15	15	0	0	
#10	1	wts	9	7	4	2	4.4	NA	NA	NA	
#11	1	wt	5	5	0	0	2.3	2.3	0	0	
#12	1	wt	9	9	0	0	2.3	2.3	0	0	
#13	3	wt	14	14	0	0	6.3	6.3	0	0	
#14	2	wt	10	10	0	0	4.6	4.6	0	0	
#15	2	wt	9	8	0	1	4.6	4.4	0	0.2	
#16	3	wt	16	16	0	0	6.9	6.9	0	0	
#17	2	wt	10	7	0	3	5.2	3.85	0.35	1	
#18	2	wt	9	9	0	0	3	3	0	0	
#19	3	wt	14	14	0	0	6.9	6.9	0	0	
#20	2	wt	11	11	0	0	5.6	5.6	0	0	
#21	1	wts	6	5	1	1	2.6	2.45	0.15	0	
#22	1	wts	7	4	4	3	3	NA	NA	NA	
#23	2	dc	8	8	0	0	4	4	0	0	
#24	2	dc	9	9	0	0	4	4	0	0	
#25	2	dc	12	12	0	0	6	6	0	0	
#26	2	dc	12	12	0	0	6	6	0	0	
#27	2	dc	10	10	0	0	4	4	0	0	
#28	2	dc	14	14	0	0	4.8	4.8	0	0	

For power lines supported by waist-type transmission towers, a total of 1.7 km of structures were misclassified (mostly false negatives), which represents 1.4% of the total studied length. If all power supply structures had been equally densely sampled, this would correspond to an accuracy of 98.6%, a precision of 99.7%, a recall of 98.9%, and an F-score of 0.993 at the point level. For power lines supported by double circuit transmission towers, a total of 740 m of structures were misclassified (false negatives only), which represents 1.4% of the total studied length and corresponds to an accuracy of 98.6%, a precision of 100%, a recall of 98.6%, and an F-score of 0.993. For power lines supported by small waist-type towers, however, the results were largely unusable with several occurrences of false positives at the tower detection stage. Tall trees at random locations were often classified as towers in such situations, which subsequently led to vegetation being classified as wires and the actual wires being missed. Overall, ≈75% of the studied length was unusable in such situations.

It was not possible provide an overall estimate of accuracy for the application of the algorithm to the entire network because the relative proportions of these three tower types was unknown. However, it was evident from our samples that waist-type towers are dominant in the studied region, with double-circuit towers also being relatively frequent and small waist-type towers relatively rare. One key factor associated with the poor results obtained for small waist-type towers was their height being sometimes lower than some of the neighbouring trees.

Imperfect classification

Figure 8A shows a subset of region #1 in which three towers are missing where the conductors split and joined back. Here, the “split towers” are of a different type to the others on the line, which explains why they were trimmed during the candidate tower identification step. We can observe that where the two side-by-side towers were missed, the section was automatically removed. This left a continuous cluster of unclassified points but none were actually misclassified. This is because the span of the connected towers that were correctly identified on each side was so large that the theoretical wire profile would be passing below ground according to the tension normally applied to this type of tower. Facing this impossible scenario, the algorithm automatically removed the entire section. Where only one of the two side-by-side towers is missing, as seen on the right hand side of Fig. 8A, the topology remains valid. In this case, only one part of the split was correctly classified while the other remained unclassified, once again without any false positive classification.

Figure 8 Subsets of datasets #1, #15 and #17 that focus on classification errors.

To make the image interpretable we chose to display only the structures belonging to the electrical network. Red points were classified as wire and blue points as transmission tower. Grey points show towers or wires that were not classified. Arrows point to the most important errors: (A) missing towers led to unclassified wires and towers, (B) a missing tower led to unclassified wires and towers, but the addition of a virtual tower led to a correct classification of the majority of the wires anyway, and (C) the sampling of the towers is so sparse that tower detection is almost impossible, and thus subsequent steps failed.

Figure 8B shows a subset of the region #15 that focuses on the missing tower. Consequently, one tower could be not matched to any other and the topology of network was interrupted. Despite this, large sections of the wires were still correctly segmented, again avoiding false positives. This was attributable to the addition of a virtual tower, as explained previously, which left only a small portion of the wires unclassified with no false positives.

Figure 8C shows the entire region #17 in which the segmentation missed a lot of features. This scene corresponds to an extreme case where facilities are so sparsely sampled that we can only see the insulators of the transmission tower in the point cloud. Consequently, several towers were not found, and thus all subsequent processes failed. Because of the missing towers, large spans were computed, and consequently catenary curves went very low. This explains why some parts of the vegetation were classified as wire. Despite this extreme case being very hard to segment, approximately 75% of the scene was still perfectly segmented. Figure 2D shows another part of this evaluation dataset that was perfectly segmented.

These three cases are representative of the errors typically found in our validation exercise. The latter example can be used to highlight the importance of reporting results in terms of the proportion of correctly classified section lengths. Indeed, due to the the lower accuracy of the method in areas with low point densities, a point-counting estimate of the overall accuracy of the algorithm would be artificially high. In this study, a point-counting estimate would have produced an accuracy value above 99.5% compared to 98.9% for a length-based assessment.

Discussion

Accuracy and scope of the study

The application of our algorithm provided excellent results with an accuracy of 98.6% (F-score = 0.993). However, the high accuracy of our method should be taken in the specific context of this study as they are not directly comparable to other methods. First, unlike previous methods cited in the introduction, the application of our method is dependent on the availability of a priori information. Second, our method applies to a totally different type of data than is analysed using most of the other existing methods. As highlighted in the introduction, line-shape-based methods would have likely failed at segmenting our data, at least where the point density is low. Conversely, our method would also fail to segment several of the cases reported in the literature. The accuracy of our results should therefore be interpreted within scope of our study i.e. the percentage of high voltage facilities that can reliably and accurately be removed from low density point clouds to facilitate the subsequent study of the neighbouring vegetation. This method was never intended to be used for monitoring the state of the electrical distribution network. We found McLaughlin (2006) and Zhu & Hyyppä (2014) to be the two closest comparable studies to this one. Zhu & Hyyppä (2014) studied power line segmentation in a forestry context and achieved an average correct classification rate of 93% with 55 pts/m2 on approximately 2.5 km of power lines. McLaughlin (2006) worked with a low density point cloud (2.5 points/m2) in a forested area testing their method on a relatively larger network (14 km) and obtained an accuracy of 82%.

Further improvements could be implemented relatively easily by refining the tower detection method in step 1, which represents a very small fraction of the source code (50 lines among more than 1,000) for the two most frequent tower types in our study area (waist-type and double-circuit). The tower detection step is indeed the most critical in the process. When tower detection reaches an accuracy of 100%, the overall segmentation will also generally reach an accuracy of 100%. However, if a single tower is missing, or worse a false positive tower is detected, the algorithm will start failing.

An important feature of the method is that, by design, the errors are grouped in contiguous clusters. This is important and highly advantageous for safely cleaning the point cloud. When false positives or false negatives can be randomly spread over the study area, even a 99% accuracy can lead to a high percentage of pixels in which the underlying vegetation is statically hidden by the remaining outliers, as explained in the introduction. Our method ensures that the vast majority of pixels will be fully cleaned and that only a minority of contiguous pixels may contain measurement artefacts.

Imperfect classification and unsupported cases

Most of the regions tested were perfectly segmented according to our manual inspection; however, some errors were still occurring when towers were badly detected. Because our method relies fully on the location of the towers to find the wires, a single error in the tower tracking step may have cascading effects. We observed four cases:Figure 8B shows a very simple case where a single tower that apparently looks easy to find was missing for an unknown reason. This case will probably be better handled as a result of the continuous improvement of the tower candidate correction step, but we think it highlights that some single towers might be missed. In such cases, the introduction of a virtual tower can limit the effect of a single missing tower on the wire detection. Our example shows that wires were mostly correctly segmented and only a small proportion were missing. Virtual towers can be seen as safeguards providing greater robustness to the method.

Figure 8A shows a case that was simply not handled by our method. We did not anticipate the existence of such topologies and we discovered this case at the validation stage. This case will be harder to process correctly but it remains a rare occurrence over the full network. Limiting errors in such cases will require in-depth redesign of the topology reconstruction step, as well as an improvement of the tower candidate correction step.

Figure 8C shows a very complex scene where the electrical facilities are only sparsely sampled. Towers are often reduced to 10 to 30 points at their very top and we actually see only the suspension insulators. Despite the limitations of the method in very low density point clouds we found a smaller proportion of false classification than expected and the segmentation accuracy still reached 75%. In the portions that were incorrectly classified, the human eye could not even distinguish the suspension insulators of the towers.

Regions #10 #23 and #24 presented cases that were badly handled where small transmission towers were situated adjacent to tall trees, which were wrongly interpreted as towers. Improving this result will require further refinement of the candidate tower correction step, which represents a very small portion of the code that drives the entire method.

In case 4, two situations may occur: (1) a tall tree is close to a tower and the local maximum filter detects the tree instead of the tower, leading to one false negative and one false positive tower; (2) several tall trees are detected as towers and even if the correction step is robust, a few false positive towers may remain because trees and towers are more similar than expected. While this has resulted in very poor outcomes, our results suggest that producing a more robust tower detection method in step 1 would be sufficient to improve the segmentation accuracy. No modifications would likely be required for the other three steps.

Case number 3 is to put in the perspective of case 4. The first step of our method could be made more robust by the addition of tests aiming to determine if the distribution of points matched the expected morphology of a transmission tower rather than another object, such as a tree. Such tests were not included in our method because of situations where the point density is so low that even the human eye is unable to recognize the features of transmission towers in the data. The reality of our sampling implied an important trade-off between the capacity of our algorithm to robustly make the distinction between a tree and a tower, and the ability to robustly recognize towers that are so sparsely sampled that their morphological features are unrecognizable. Our method was made relatively robust to sparse sampling but this comes at the expense of the detectability of small towers. The threshold density at which power supplies are no longer recognizable is hard to estimate. Dataset #17 was sampled with a nominal density of 2.6 points/m2, but what matters is the sampling density of the target structures, which may also be affected by the direction of the moving sensor relative to the structure. In this case it was both a low-density point cloud and a sensor moving perpendicular to the power lines.

There are obviously some other limitations to our algorithm for which consequences were not observed in our dataset. Our inability to fully anticipate all possible features of high-tension power lines implies that we cannot provide an exhaustive list, but the following limitations are worth mentioning. Our method does not apply to networks supported by different types of transmission towers along the same wires. A typical case can be found in Chen et al. (2018) figures 10 and 11. Similarly, it does not apply to networks with very close side-by-side power supply facilities of different types e.g. one row of waist-type towers adjacent to another row of double-circuit towers. Also, any overly-complex scenes, such as networks located in the vicinity of generators containing several wires and towers, would likely make the method inapplicable.

Priors and limits of application

Our method is assisted by prior knowledge of the network including wire orientation and tower types. This is an obvious limitation of the method if such prior information is not available. However, high voltage power lines are usually mapped. In more complex contexts, such as in urban regions where lower voltage conductors are present, having this kind of prior knowledge may not be possible. The segmentation of urban electrical facilities requires different methods and point clouds sampled with higher densities. Our method would not be applicable in such contexts.

Nevertheless, our method still uses a limited set of priors. For example, it does not require prior knowledge of the number of power line rows. Yet, we observed that it performs well on one to four rows (four rows not shown) and there is no hard upper limit. Our method also does not require any prior knowledge of the position of the towers along the network, since it is capable of detecting them. In practice, power supply companies typically maintain maps of high-voltage transmission towers, which may lead to even more accurate results. The method also includes safeguards to prevent against irrelevant false positive classifications. This is what we showed in Fig. 8A where the ground was initially classified as wire but was automatically corrected, and in Fig. 8B where a virtual tower prevented against missing too many wire points. Also, unlike previous methods presented in the literature, we proved that our method works not only on linear sections, but also supports branching and deflections of networks. It works with partial objects in low-density point clouds and does not need any special flying pattern (e.g. following the power line). We believe it is thus applicable to any natural landscape in which broad ALS coverage is available.

Implementation and reproducible work

Broad ALS coverage is normally split into multiple files. We presented the method on single blocks of data loaded in memory, but in practice it would not be possible to load thousands of tiles at once. Unlike methods presented in the literature that require pre-processing to fit each segment into a single file (e.g. Guo et al., 2019), our method works independently of the tiling pattern. Each tile is loaded with a buffer, thus ensuring that the virtual towers that are added at its edges are actually out of the processed region and do not affect the classification. This is made possible through the file collection processing engine provided with the lidR package (Roussel & Auty, 2021; Roussel et al., 2020). The method is therefore not only workable on small samples but it is also fully integrated into a framework dedicated to processing large areas seamlessly.

The text presented in this paper would not allow the methods to be fully reproduced. We have presented the overall concept, but for the sake of concision several details were omitted about candidate tower cleaning, virtual towers and deflection towers, for example. To ensure reproducibility, the source code has been made publicly available, so that any interested user can implement the method and analyse it in detail. The implementation is made of four functions corresponding to the four steps described in the materials and methods section. Any step can thus be replaced easily by another method to match users’ needs without affecting the other steps.

The method is available within the R language (R Core Team, 2020) and is implemented in the lidRplugins package available at https://github.com/Jean-Romain/lidRplugins. lidRplugins is an extension of the lidR package (Roussel & Auty, 2021; Roussel et al., 2020) and contains experimental methods. Depending on its future improvements and success when applied to other datasets, this algorithm might at some stage be made directly available in the lidR package.

The package does not record every existing transmission tower type in a database. Instead, users have the capability to define and use their own tower types, which makes the method reproducible in jurisdictions where power lines are supported by tower types that are different to those presented here.

Conclusions

We created an algorithm dedicated to segmenting electrical network facilities in ALS data over vast natural landscapes. The method is designed to segment transmission towers, conductor wires and earth wires from high-voltage power lines. The method was developed to clean point clouds in an attempt to facilitate ecology- and forestry-oriented studies that use ALS data. Our approach relies on prior information about the network and is thus not fully unsupervised. It is, however, applicable to vast areas containing a wide range of specific cases not explicitly addressed in previous studies, such as wire splitting, wire forking, complex topography, complex topology, earth wires, partial wire sampling, partial tower sampling and non-optimized sampling for electrical network surveys. We studied the limitations of the method through an application in northeastern Quebec, Canada.

From this analysis we conclude that limiting false positives or false negatives at the tower detection stage (step 1) is key to the success of the segmentation. We highlighted the limitations of our methods, but over the whole network tower detection errors remained a rare occurrence. It may be possible to refine the tower candidate correction step with minor efforts and without redesigning the whole algorithm, which has proven to be robust. Further development efforts could be dedicated to the automatic recognition of the tower types to support some cases not covered by our methods, such as power lines supported by various tower types. However, this would only be possible for datasets in which the point density is sufficient to make the morphological features of transmission towers recognizable. Finally, to ensure reproducibility and allow further development, we also provided a ready-to-use open-source tool implemented in a framework dedicated to processing vast areas.

Additional Information and Declarations

Competing Interests

Author Contributions

Data Availability

The authors declare that they have no competing interests.

Jean-Romain Roussel conceived and designed the experiments, performed the experiments, analyzed the data, performed the computation work, prepared figures and/or tables, authored or reviewed drafts of the paper, and approved the final draft.

Alexis Achim conceived and designed the experiments, authored or reviewed drafts of the paper, and approved the final draft.

David Auty conceived and designed the experiments, authored or reviewed drafts of the paper, and approved the final draft.

The following information was supplied regarding data availability:

The source code and example data are available at GitHub: https://github.com/Jean-Romain/lidRplugins

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
