# Peer review of "Classification of high-voltage power line structures in low density ALS data acquired over broad non-urban areas"

_PeerJ Computer Science, doi:10.7717/peerj-cs.672_

## Round 0.1 · original submission · Major Revisions

Please carefully address all comments from reviewers, particularly regarding innovations and validations. There are also some presentations issues which should be improved as well.

Reviewer 1 ·

Basic reporting

The literature is not sufficient. For example, line number 53-57 and 65-66 do mention "some studies". Which studies? What are the gaps which motivated you to conduct this research?

Introduction is but messy--Reorganise it. The authors has tried to provide gaps to highlight motivation but the discussion is too divergent. Why solving this problem is really important? There are few recent studies that you have missed to include.

What outcome is actually expected?

Experimental design

The research gaps are not well presented. What authors want to achieve is not well stated. Section 2, provides all details in one section briefly. For example, if you discussed about datasets, why not to provide more details about it (especially, reliability concerns about the dataset). Validation, which should be one of the main focus, is presented briefly.
Results and discussion is okay but they are not well connected with previous and further sections. Elaborate each section and make a link between them.
Conclusion need to be modified--what is the key take away message? Do your findings has applicability in industries and how it could provide value to them?

Validity of the findings

The validation is provided but there are many threats to validity, which are not addressed. Provide the details about different threats and mention that how you actually addressed them (if you have) or failed to address. Can you study be treated as valid and accurate? The discussion is not sufficient to build this trust. You must explicitly mention it in detail.

Additional comments

I hope that incorporating my feedback and restructuring your article, will help to improve your research submission.

Reviewer 2 ·

Basic reporting

The paper has deficiencies in English expression, sentence structure, and tense usage. Please have the article proofread by an expert.

Literature is not systematically reviewed and presented in line with current research. Please add a more in depth introduction and literature review by studying more recent publications and explain what this research conveys differently.

Experimental design

The authors are advised to include main contributions or research questions to better highlight the novelty of the research.

The motivation to carry out this work is missing. The introduction section should include a discussion on this aspect.

In the present form it in very difficult to understand various equations. The authors must elaborate different variables or parameters in each equation before writing it.

The authors must include a discussion on why the eigenvalue decomposition method is used. Are there other methods available? Can they be used? Why or why not?

Para line 37-44: the authors need to establish the strong background for this claim. Some recent work along with the challenges should be included. Is it one of these challenges? Establish this with support of similar work references in this regard.

Para line 46-51: Similarly, authors need to support this claim with recent work.

Para line 53-57: References required.

Validity of the findings

Authors have included an elaborative discussion on validation of their work.

The missing aspect is:
Compare this research's findings with recent state-of-the-art and explain the benefits of the approach

Additional comments

The authors are advised to include main contributions or research questions to better highlight the novelty of the research.

The motivation to carry out this work is missing. The introduction section should include a discussion on this aspect.

In the present form it in very difficult to understand various equations. The authors must elaborate different variables or parameters in each equation before writing it.

The authors must include a discussion on why the eigenvalue decomposition method is used. Are there other methods available? Can they be used? Why or why not?

Para line 37-44: the authors need to establish the strong background for this claim. Some recent work along with the challenges should be included. Is it one of these challenges? Establish this with support of similar work references in this regard.

Para line 46-51: Similarly, authors need to support this claim with recent work.

Para line 53-57: References required.

In the present form, the literature reported is very poor. Authors are advised to include a full dedicated section on latest literature. They should also mention which research gaps they are addressing.

The paper has deficiencies in English expression, sentence structure, and tense usage. Please have the article proofread by an expert.

Literature is not systematically reviewed and presented in line with current research. Please add a more in depth introduction and literature review by studying more recent publications and explain what this research conveys differently.

Compare this research's findings with recent state-of-the-art and explain the benefits of the approach

Reviewer 3 ·

Basic reporting

This manuscript developed an algorithm for the segmentation of electrical network facilities from ALS point clouds. The English language should be improved to ensure that others could clearly understand your text. So, you should use more short sentences instead of inserting too many long ones. The literature review in Section 1 Introduction needs to be more logical. Thank you for providing the source code or tools to allow this methods to be reproduced easily.

Experimental design

This manuscript should clearly define the research question, which must be relevant and meaningful. The knowledge gap being investigated should be identified. The designed experiment is somewhat simple, and more data and results in the experimental area are not presented.The results of power line structure classification based on ALS mainly carry out qualitative analysis, which lacks the rigor of quantitative analysis.In addition, this method also has the defect of relying more on prior knowledge and manual judgment.

Validity of the findings

The experimental data should be robust, statistically sound, and controlled. The conclusions should be appropriately stated, should be connected to the original question investigated, and should be limited to those supported by the results. The method and its meaning in this study should be more clear.

Additional comments

Some issues or questions could be addressed:
1) How to distinguish between 'good' and 'poor' classification results? What are the quantitative criteria?
2) Line 190, the sentence of 'An idealised hanging chain ... assumed ... follows ...' should be clear.
3) Line 287, 'range form ...' should be 'range from ...'.
It is recommended to check the language spelling through the whole manuscript.

---

## Round 0.2 · accepted · Accept

All questions and concerns have been addressed. The reviewer committee recommends accepting the paper.

Reviewer 3 ·

Basic reporting

no comment

Experimental design

no comment

Validity of the findings

no comment

Additional comments

I have no other questions.

---

## Author Rebuttal · Round 0.2

# 1 Reviewer 1

Basic reporting

**Comment 1:** The literature is not sufficient. For example, line number 53-57 and 65-66 do mention "some studies". Which studies? What are the gaps which motivated you to conduct this research?

**Answer 1:** The introduction has entirely been rewritten (except the first paragraph). We addressed the three reviewers comments about introduction, which all converge. We have included more references and we carefully explained what motivated us to conduct this study as well as the expected outcomes.

**Change made:** Introduction entirely re-written

**Comment 2:** Introduction is but messy–Reorganise it. The authors has tried to provide gaps to highlight motivation but the discussion is too divergent. Why solving this problem is really important? There are few recent studies that you have missed to include.

**Answer 2:** See comment 1

**Comment 3:** What outcome is actually expected?

**Answer 3:** See comment 1

## 1.1 Experimental design

**Comment 4:** The research gaps are not well presented. What authors want to achieve is not well stated. Section 2, provides all details in one section briefly. For example, if you discussed about datasets, why not to provide more details about it (especially, reliability concerns about the dataset). Validation, which should be one of the main focus, is presented briefly.

**Answer 4:** The research gap and what we want to achieve is now better presented in the introduction. Following the suggestions of the reviewers, the validation has been entirely modified and strengthened. We provide an improved explanation of the reason why a validation based on the percentage of correctly or incorrectly classified points would not be relevant in our study and we have chosen to validate our method based on the length of correctly and incorrectly classified wire sections. This is now explained in the materials and methods. While the method is not standard in the literature, we think the explanations will be convincing. We wish to thank the reviewer for highlighting this issue with our validation. In line with this suggestion, we now report accuracy, precision, recall, and F-score from our validation.

**Change made:** Introduction entirely re-written, new validation method and metrics

**Comment 5:** Results and discussion is okay but they are not well connected with previous and further sections. Elaborate each section and make a link between them.

**Answer 5:** The introduction, results and discussion sections were reworked substantially

**Change made:** Substantial amendments made to the discussion, mainly in the first two sections.

**Comment 6:** Conclusion need to be modified–what is the key take away message? Do your findings has applicability in industries and how it could provide value to them?

**Answer 6:** With the new introduction and discussion, we believe part of this issue with the take-away message will be fixed. One important element was to focus more clearly on our main goal, which unlike most previous studies was not oriented towards network monitoring, but instead to remove the points to clean the scene and facilitate the study of the vegetation. In addition to this, we modified the conclusion as recommended.

**Change made:** Conclusion modified to include a clear take-away message on the important of the tower detection step.

## 1.2 Validity of the findings

**Comment 7:** The validation is provided but there are many threats to validity, which are not addressed. Provide the details about different threats and mention that how you actually addressed them (if you have) or failed to address. Can you study be treated as valid and accurate? The discussion is not sufficient to build this trust. You must explicitly mention it in detail.

**Answer 7:** We believe that our answer to comment 4 addressed this comment. The validation approach has been entirely modified in line with the suggestions. We also provided a better justification for the chosen approach.

## 1.3 Comments for the Author

**Comment 8:** I hope that incorporating my feedback and restructuring your article, will help to improve your research submission.

**Answer 8:** It did for sure. Thank you for your constructive review and comments.

# 2 Reviewer 2

## 2.1 Basic reporting

**Comment 9:** The paper has deficiencies in English expression, sentence structure, and tense usage. Please have the article proofread by an expert.

**Answer 9:** The paper has been reviewed by a native English (British) speaker who is very familiar with scientific writing.

**Comment 10:** Literature is not systematically reviewed and presented in line with current research. Please add a more in depth introduction and literature review by studying more recent publications and explain what this research conveys differently.

**Answer 10:** The introduction has been entirely rewritten (except the first paragraph). We addressed the three reviewers' comments about the introduction, which all converge, so we now think the manuscript is much more coherent. We included more references to better position our work with regards to the most up-to-date knowledge on this topic, and carefully explained the specificity of our study. Through this process, we believe we were able to better present the knowledge gap, and thus the main contribution and novelty of our work.

**Change made:** Introduction entirely re-written

## 2.2 Experimental design

**Comment 11:** The authors are advised to include main contributions or research questions to better highlight the novelty of the research.

**Answer 11:** See answer to comment 10

**Comment 12:** The motivation to carry out this work is missing. The introduction section should include a discussion on this aspect.

**Answer 12:** See answer to comment 10

**Comment 13:** In the present form it in very difficult to understand various equations. The authors must elaborate different variables or parameters in each equation before writing it.

**Answer 13:** There is a single equation. We do not understand to which "various" equations the reviewer is referring to. We do agree that the catenary equation is relatively complex. This is why we split it into 3 pieces so each piece fits a single line. This way, we believe it is a little easier to understand than the form presented in the cited paper. A key point here is that understanding the equation is not necessary to understand the paper. We added because it is a key part of the algorithm. One advantage is that it saves interested readers the need to access and read the full paper of Hatibovic 2014, which contains several similar complex equations that

are often displayed over several lines.

**Comment 14:** The authors must include a discussion on why the eigenvalue decomposition method is used. Are there other methods available? Can they be used? Why or why not?

**Answer 14:** We did not use eigenvalues decomposition. It is a commonly used approach for wire detection. We did explain how its use may be considered in high-density point clouds but not in low-density ones.

**Change made:** Newly presented introduction that includes a better description of the two main families of methods with their reported accuracy, and of the shortcomings in the context of this study L77-94

**Comment 15:** Para line 37-44: the authors need to establish the strong background for this claim. Some recent work along with the challenges should be included. Is it one of these challenges? Establish this with support of similar work references in this regard.

**Answer 15:** This paragraph has been rewritten to explain how ALS data are processed in forestry and ecology and consequently how human-made structures can affect local biomass predictions
.

**Change made:** Lines 37-44 were entirely re-written in line with these comments.

**Comment 16:** Para line 46-51: Similarly, authors need to support this claim with recent work.

**Answer 16:** We cannot support this claim with the literature. Masking spatial data with a buffer is a very common procedure when processing geospatial data. Authors usually do not report this step as it is just common sense. Biomass predictions are locally (wildly) erroneous because of power lines. In the absence of other solutions, they buffer out the data using geospatial tools from GIS software.

**Change made:** We removed this paragraph that did not provide much additional information. Instead, we improved the description of the knowledge gap and of the reason for developing this algorithm, which is to allow the study of the underlying and neighbouring vegetation.

**Comment 17:** Para line 53-57: References required.

**Answer 17:** We agree that these were unsupported facts. We believe the new introduction now provides better context.

**Change made:** In the process of re-writing the introduction, this paragraph was removed as it did not bring much useful information.

## 2.3    Validity of the findings

**Comment 18:** Authors have included an elaborative discussion on validation of their work.

**Answer 18:** See comment 19

**Comment 19:** The missing aspect is: Compare this research's findings with recent state-of-the-art and explain the benefits of the approach

**Answer 19:** We added some important information in our revised manuscript to put the work in the context of previous studies. A new validation has been produced (see comment 4) and consequently some paragraphs were changed/moved/added/deleted in the results and discussion. We believe the new version addresses this comment.

**Change made:** Introduction re-written, validation done in a different way and discussion amended. See in particular the second paragraph of the new discussion L77-94.

## 2.4    Comments for the Author

**Comment 20:** The authors are advised to include main contributions or research questions to better highlight the novelty of the research.

**Answer 20:** See comment 10

**Comment 21:** The motivation to carry out this work is missing. The introduction section should include a discussion on this aspect.

**Answer 21:** See comment 10

**Comment 22:** In the present form it in very difficult to understand various equations. The authors must elaborate different variables or parameters in each equation before writing it.

**Answer 22:** See comment 13

**Comment 23:** The authors must include a discussion on why the eigenvalue decomposition method is used. Are there other methods available? Can they be used? Why or why not?

**Answer 23:** See comment 14

**Comment 24:** Para line 37-44: the authors need to establish the strong background for this claim. Some recent work along with the challenges should be included. Is it one of these challenges? Establish this with support of similar work references in this regard.

**Answer 24:** See comment 15

**Comment 25:** Para line 46-51: Similarly, authors need to support this claim with recent work.

**Answer 25:** See comment 16

**Comment 26:** Para line 53-57: References required.

**Answer 26:** See comment 17

**Comment 27:** In the present form, the literature reported is very poor. Authors are advised to include a full dedicated section on latest literature. They should also mention which research gaps they are addressing.

**Answer 27:** See comment 10

**Comment 28:** The paper has deficiencies in English expression, sentence structure, and tense usage. Please have the article proofread by an expert.

**Answer 28:** See comment 9

**Comment 29:** Literature is not systematically reviewed and presented in line with current research. Please add a more in depth introduction and literature review by studying more recent publications and explain what this research conveys differently.

**Answer 29:** See comment 10

**Comment 30:** Compare this research's findings with recent state-of-the-art and explain the benefits of the approach

**Answer 30:** See comments 4,10 and 20

# 3 Reviewer 3

## 3.1 Basic reporting

**Comment 31:** This manuscript developed an algorithm for the segmentation of electrical network facilities from ALS point clouds. The English language should be improved to ensure that others could clearly understand your text. So, you should use more short sentences instead of inserting too many long ones. The literature review in Section 1 Introduction needs to be more logical. Thank you for providing the source code or tools to allow this methods to be reproduced easily.

**Answer 31:** The paper has been reviewed by a native English (British) speaker who is very familiar with scientific writing. The introduction has been entirely rewritten. Many thanks for highlighting the importance of sharing the source code. We believe this is a key benefit of this study. See also our answer to comment 1.

**Change made:** Introduction entirely re-written

## 3.2 Experimental design

**Comment 32:** This manuscript should clearly define the research question, which must be relevant and meaningful. The knowledge gap being investigated should be identified. The designed experiment is somewhat simple, and more data and results in the experimental area are not presented.The results of power line structure classification based on ALS mainly carry out qualitative analysis, which lacks the rigor of quantitative analysis. In addition, this method also has the defect of relying more on prior knowledge and manual judgment.

**Answer 32:** See comment 4

**Change made:** Introduction entirely re-written. Also, we have now provided a much more detailed, quantitative analysis of the results. We believe this rigorous quantitative analysis strongly improves the manuscript. Many thanks for this comment.

## 3.3 Validity of the findings

**Comment 33:** The experimental data should be robust, statistically sound, and controlled. The conclusions should be appropriately stated, should be connected to the original question investigated, and should be limited to those supported by the results. The method and its meaning in this study should be more clear.

**Answer 33:** Introduction was re-written to better state the problem we are trying to solve, the validation was reworked (see comment 4), the discussion was improved to better focus and better connect with the introduction and results. Through all these major changes, we believe that we have addressed this comment.

**Change made:** Introduction entirely re-written, validation was re-done in a different way. The metric used in the validation is also better justified.

## 3.4 Comments for the Author

Some issues or questions could be addressed:

**Comment 34:** 1) How to distinguish between 'good' and 'poor' classification results? What are the quantitative criteria?

**Answer 34:** The new validation is entirely quantitative, so this comment no longer applies.

**Comment 35:** 2) Line 190, the sentence of 'An idealised hanging chain ... assumed ... follows ...' should be clear.

**Answer 35:** We are not sure we understand the problem. The sentence was reviewed and modified by a native English speaker.

**Change made:** Sentence changed

**Comment 36:** 3) Line 287, 'range form ...' should be 'range from ...'.

**Change made:** Fixed thank you

**Comment 37:** It is recommended to check the language spelling through the whole manuscript.

**Answer 36:** See comment 31